# Study of Reducing Atmospheric Turbulence-Induced Beam Wander of a Twisted Electromagnetic Elliptical Vortex Beam

**Kai Huang [1,2], Yonggen Xu [3,\*], Yuqiang Li [4] and Jin Cao [1,4]**

1   College of Mathematics and Physics, Leshan Normal University, Leshan 614000, China; hk@lsnu.edu.cn (K.H.); caojin@lsnu.edu.cn (J.C.)
2   Key Laboratory of Detection and Application of Space Effect in Southwest Sichuan, Leshan Normal University, Leshan 614000, China
3   Department of Physics, School of Science, Xihua University, Chengdu 610039, China
4   Group of Applied Astronomy Research, Yunnan Observatory, Chinese Academy of Sciences, Kunming 650011, China; lyq@ynao.ac.cn
\*   Correspondence: xuyg@mail.xhu.edu.cn

**Abstract:** We derive the analytical expressions for root-mean-square (rms) beam wander (BW) and relative BW of a twisted electromagnetic elliptical vortex (TEEV) beam propagating through non-Kolmogorov atmospheric turbulence with the help of the extended Huygens–Fresnel principle and the second-order moments of the Wigner distribution function (WDF). Our numerical findings demonstrate that the BW of a TEEV beam with a small ellipticity, a large topological charge as well as a small waist width and initial coherent length is less affected by the turbulence. It can be also found that the effect of turbulence with a larger outer scale of turbulence, a generalized exponent parameter, and a generalized structure parameter on BW is more obvious. It is interesting to find that the effect of atmospheric turbulence on BW for a TEEV beam can be effectively reduced by regulating jointly the symbols and sizes of the twisted factor and topological charge. Therefore, modulation of the structure parameters of a TEEV beam provides a new way to mitigate turbulence-induced beam wander. Our work will be useful for free-space optical communications, remote sensing, and lidar distance measurement.

**Keywords:** twisted electromagnetic elliptical vortex beam; non-Kolmogorov atmospheric turbulence; beam wander

## 1. Introduction

It is well known that atmospheric turbulence will cause a series of deleterious turbulence effects, such as beam spreading and beam wander, scintillation (i.e., intensity fluctuation), and phase distortion, which will result in an increase in the communication error rate and a drop in the channel capacity of the optical system [1–10]. It has been experimentally investigated that general atmospheric turbulence possesses a turbulence structure different from the typical Kolmogorov's one. It could have other energy distributions among differently-sized turbulent eddies and exhibit non-homogeneity, i.e., the non-Kolmogorov turbulence [11–17]. Over the past several decades, many works have been carried out concerning the propagation of various laser beams in atmospheric turbulence due to its important applications in free-space optical communications, remote sensing, and lidar distance measurement [8–20]. Beam wander (BW) is characterized by the random displacement of the instantaneous center of a laser beam as it propagates through atmospheric turbulence, which is an important property determining the performance of laser beams in practical applications, such as beam tracking and pointing, in high-energy laser systems and military applications [4,7,16–18]. BW is a significant limitation in the above applications; therefore, it is of practical importance to develop some methods to overcome

or alleviate the turbulence-induced BW, and it is also revealed in [21–40] that one-way use of a laser beam with special spatial structures can achieve this goal.

A partially coherent (PC) beam is less affected by atmospheric turbulence than a fully coherent beam [21–25] and can effectively lighten the speckle and reduce beam wander [7,8,16–20]. In 1992, Allen et al. found that each photon in a laser beam with a phase factor of exp(imθ) has an orbital angular momentum (OAM) of ℏm, where ℏ is the reduced Planck constant and m is the topological charge carried by the beam. This kind of PC beam with OAM is called a partially coherent vortex (PCV) beam [26–30]. It has been demonstrated that a PCV beam and a partially coherent elliptical vortex (PCEV) beam have a stronger anti-turbulence ability propagating through atmospheric turbulence over a non-vortex beam [13,14,20,31–33]. We also find that, in optical information transmission, the advantage of a PCV beam is that OAM coding or OAM multiplexing can be used to extremely improve the channel capacity of a communication system [26–33]. In addition to the vortex beam, there is another nontrivial beam (named a twisted beam) that induces the beam carrying OAM [20,34–40]. It is shown that a partially coherent twisted (PCT) beam not only induces the rotation of a laser beam on propagation, but also has advantages in resisting turbulence-induced negative effects, such as scintillation, degeneration, depolarization, and overcoming the classical Rayleigh limit and information transfer through atmospheric turbulence [34,35,40,41].

In this paper, we combine the twisted phase, vortex phase, ellipticity, and coherence of a laser beam, so a new kind of twisted electromagnetic elliptical vortex (TEEV) beam is introduced. The analytical expressions for root-mean-square (rms) BW and relative BW of the TEEV beam propagating through non-Kolmogorov atmospheric turbulence with the help of the extended Huygens–Fresnel principle are derived, and the effects of the twisted factor, topological charge, ellipticity, and coherent length on the rms BW and relative BW are explored in detail. Our strategy provides a new way to reduce turbulence-induced BW and promote the application of a TEEV beam in free-space optical communications, remote sensing, and lidar distance measurement.

## 2. BW of a TPCEV Beam in Non-Kolmogorov Atmospheric Turbulence

The BW of a fully coherent (FC) beam can be described statistically as the variance of the displacement of the instantaneous center of the beam propagating through a turbulent atmosphere. A model of BW of an FC beam valid under general poor turbulence conditions has been proposed by Andrews and Phillips [4], and in 2012, the theory model was extended to the case of PC beams [16–18]:

$$
\begin{aligned}
\left\langle r_c^2 \right\rangle = 4\pi^2 k^2 W_{FS}^2 \int_0^z \int_0^\infty \kappa \Phi_n(\kappa) \exp\left(-\kappa^2 W_{LT}^2\right) \\
\times \left\{1 - \exp\left[-\frac{2z^2\kappa^2(1 - z'/z)^2}{k^2 W_{FS}^2}\right]\right\} d\kappa dz',
\end{aligned}
\tag{1}
$$

where, $k = 2\pi/\lambda$ is the wave number with $\lambda$ being the wavelength of the PC beam, $z$ denotes the total propagation path length, $z'$ is the distance of an intercept point from the input plane at $z = 0$, $\Phi_n(\kappa)$ is the spatial power spectrum of the refractive index fluctuations of the non-Kolmogorov atmospheric turbulence, and $\kappa$ is the magnitude of the spatial wavenumber. $W_{LT}$ and $W_{FS}$ are the long-term beam widths at the receiver plane in the presence of turbulence and in the free space, respectively. To model atmospheric turbulence, the form of the non-Kolmogorov turbulence spectrum can be expressed as follows [4,11–17]:

$$
\Phi_n(\kappa, \alpha) = \frac{A(\alpha) C_n^2 \exp\left[-(\kappa^2/\kappa_m^2)\right]}{(\kappa^2 + \kappa_0^2)^{\alpha/2}}, \quad 0 \le \kappa < \infty, \; 3 < \alpha < 4,
\tag{2}
$$

where, $C_n^2$ is the generalized structure parameter with units $m^{3-\alpha}$, $\alpha$ denotes the generalized exponent parameter, $\kappa_m = c(\alpha)/l_0$ with $l_0$ being the inner scale of turbulence, and $\kappa_0 =$

$2\pi/L_0$ with $L_0$ being the outer scale of turbulence. $A(\alpha)$ and $c(\alpha)$ are given by the following [11–17]:

$$A(\alpha) = \frac{1}{4\pi^2}\Gamma(\alpha-1)\cos\left(\frac{\alpha\pi}{2}\right) \tag{3}$$

$$c(\alpha) = \left[\frac{2\pi}{3}\Gamma\left(\frac{5-\alpha}{2}\right)A(\alpha)\right]^{\frac{1}{\alpha-5}} \tag{4}$$

where, $\Gamma(\cdot)$ is the Gamma function. When $\alpha = 11/3$, we can calculate $A(11/3) = 0.033$, $c(11/3) = 5.91$, and $\kappa_m = 5.91/l_0$. Equation (2) is the von Karman spectrum. When $\alpha = 11/3$ and $L_0 = \infty$, Equation (2) reduces to the Tatarskii spectrum; furthermore, the spectrum expressed in Equation (2) also reduces to the conventional Kolmogorov spectrum when $\alpha = 11/3$, $L_0 = \infty$ and $l_0 = 0$ [4,16–18].

For the sake of a better understanding of the BW, it is helpful to simplify Equation (1) by applying the geometrical optics approximation: $1 - \exp[-2z^2\kappa^2(1 - z'/z)^2/(k^2W_{FS}^2)] \approx 2z^2\kappa^2(1 - z'/z)^2/(k^2W_{FS}^2)$, where, $z\kappa^2 \ll k$ and the diffraction effect is neglected [4,16–20]. Therefore, the BW of a PC beam can be expressed approximately by the following:

$$\begin{aligned}
\left\langle r_c^2 \right\rangle = {} & \frac{4\pi^2 C_n^2 A(\alpha)\kappa_0^{-\alpha}z^2}{(\alpha-2)} \\
& \times \int_0^z \left(1-\frac{z'}{z}\right)^2 \left\{-2\kappa_0^4 + \kappa_0^\alpha\kappa_m^2\left(W_{LT}^2 + \kappa_m^{-2}\right)^{\alpha/2}\right. \\
& \times \left[2\kappa_0^2\left(1+\kappa_m^2 W_{LT}^2\right)+(\alpha-2)\kappa_m^2\right] \\
& \times \left(1+\kappa_m^2 W_{LT}^2\right)^{-2}\exp\left(\kappa_0^2/\kappa_m^2+\kappa_0^2 W_{LT}^2\right) \\
& \left.\times \Gamma\left[2-\frac{\alpha}{2},\left(\kappa_0^2/\kappa_m^2+\kappa_0^2 W_{LT}^2\right)\right]\right\} dz',
\end{aligned} \tag{5}$$

where $\Gamma(\cdot,\cdot)$ represents the incomplete Gamma function. Equation (5) is a general formula of BW for a random PC beam propagating through non-Kolmogorov atmospheric turbulence. It can be easily found from Equation (5) that the BW of a PC beam is determined by the $W_{LT}$ and turbulence parameters. According to Refs.[4,16–20], $W_{LT}$ was calculated by the method of second-order moments and can depend on the initial parameters of a PC beam.

Therefore, in this paper, in order to study and reduce turbulence-induced BW, a new kind of twisted electromagnetic elliptical vortex (TEEV) beam is introduced as well as a 2 × 2 cross-spectral density matrix (CSDM) [18–25]

$$\overleftrightarrow{W}(\boldsymbol{\rho}_{01},\boldsymbol{\rho}_{02};0)=\begin{bmatrix} W_{xx}(\boldsymbol{\rho}_{01},\boldsymbol{\rho}_{02};0) & W_{xy}(\boldsymbol{\rho}_{01},\boldsymbol{\rho}_{02};0) \\ W_{yx}(\boldsymbol{\rho}_{01},\boldsymbol{\rho}_{02};0) & W_{yy}(\boldsymbol{\rho}_{01},\boldsymbol{\rho}_{02};0) \end{bmatrix}, \tag{6}$$

where, $\boldsymbol{\rho}_{01} \equiv (x_{01}, y_{01})$ and $\boldsymbol{\rho}_{02} \equiv (x_{02}, y_{02})$ are the traverse vectors of two arbitrary points in the incident plane. $W_{pq}(\boldsymbol{\rho}_{01}, \boldsymbol{\rho}_{02}) = \langle E_p^*(\boldsymbol{\rho}_{01})E_q(\boldsymbol{\rho}_{02})\rangle$, $(p,q = x,y)$, $E_x$ and $E_y$ are the two fluctuating components of the stochastic electric vector with respect to two mutually orthogonal $x$ and $y$ directions [10,14–16,39]. Where the angular brackets $\langle\cdots\rangle$ denote the ensemble average and the asterisk denotes the complex conjugate. It is striking to note that the two mutually orthogonal components $E_x$ and $E_y$ are uncorrelated at each point of the source. The off-diagonal elements of the CSDM in Equation (6) for the TEEV beam in the source plane $W_{xy}(\boldsymbol{\rho}_{01}, \boldsymbol{\rho}_{02}; 0)$ and $W_{yx}(\boldsymbol{\rho}_{01}, \boldsymbol{\rho}_{02}; 0)$ are set to zero. Therefore, the diagonal elements of

the CSDM in Equation (6) $W_{xx}(\rho_{01}, \rho_{02}; 0)$ and $W_{yy}(\rho_{01}, \rho_{02}; 0)$ can be expressed as follows [10,14–16,39]:

$$
\begin{aligned}
W_{pp}(\boldsymbol{\rho}_{01}, \boldsymbol{\rho}_{02}; 0) = & \left( \frac{x_{01}}{w_{0x}} - i\frac{y_{01}}{w_{0y}} \right)^m \left( \frac{x_{02}}{w_{0x}} + i\frac{y_{02}}{w_{0y}} \right)^m \\
& \times \exp\left( -\frac{x_{01}^2 + x_{02}^2}{w_{0x}^2} - \frac{y_{01}^2 + y_{02}^2}{w_{0y}^2} \right) \\
& \times \exp\left[ -\frac{(x_{01} - x_{02})^2 + (y_{01} - y_{02})^2}{2\delta_{pp}^2} \right] \\
& \times \exp\left[ -ik\mu(x_{01}y_{02} - x_{02}y_{01}) \right] \\
& p = x, y,
\end{aligned}
\tag{7}
$$

where $w_{0x}$ and $w_{0y}$ are the beam waist widths of the TEEV beam in the $x$ and $y$ directions, respectively. $m$ and $\mu$ denote the topological charge and twist factor of the TEEV beam, respectively. $\delta_{xx}$ and $\delta_{yy}$ represent the initial transverse coherent lengths of the TEEV beam along the $x$ and $y$ directions, respectively. The trace of the CSDM of the TEEV beam in the source plane $z = 0$ is expressed as the following [18–25]:

$$
W_{tr}(\boldsymbol{\rho}_{01}, \boldsymbol{\rho}_{02}; 0) = Tr\left[ \overrightarrow{W}(\boldsymbol{\rho}_{01}, \boldsymbol{\rho}_{02}; 0) \right] = W_{xx}(\boldsymbol{\rho}_{01}, \boldsymbol{\rho}_{02}; 0) + W_{yy}(\boldsymbol{\rho}_{01}, \boldsymbol{\rho}_{02}; 0)
\tag{8}
$$

where $Tr$ denotes the trace of the CSDM of the TEEV beam. The propagation of the trace of the CSDM of the TEEV beam in non-Kolmogorov turbulence can be given as follows [11–20]:

$$
\begin{aligned}
W_{tr}(\boldsymbol{\rho}, \boldsymbol{\rho}_d; z) = & \frac{k^2}{4\pi^2 z^2} \iint W_{tr}(\boldsymbol{\rho}_0, \boldsymbol{\rho}_{0d}; 0) \\
& \times \exp\left\{ \frac{ik}{z}[(\boldsymbol{\rho} - \boldsymbol{\rho}_0) \cdot (\boldsymbol{\rho}_d - \boldsymbol{\rho}_{0d})] - \frac{1}{2}D_w(\boldsymbol{\rho}_d, \boldsymbol{\rho}_{0d}; z) \right\} d^2\boldsymbol{\rho}_0 d^2\boldsymbol{\rho}_{0d}.
\end{aligned}
\tag{9}
$$

where $z$ denotes the propagation distance. The two-point spherical wave structure function $D_w(\rho_d, \rho_{0d}; z)$ is expressed as follows:

$$
D_w(\boldsymbol{\rho}_d, \boldsymbol{\rho}_{0d}; z) = 8\pi^2 k^2 z \int_0^1 d\xi \int_0^\infty [1 - J_0(\kappa|\xi\boldsymbol{\rho}_{0d} + (1-\xi)\boldsymbol{\rho}_d|)]\Phi_n(\kappa)\kappa d\kappa,
\tag{10}
$$

where $J_0(\cdot)$ denotes the Bessel function of the first kind and zero order. We use the central abscissa coordinate systems, that is,

$$
\boldsymbol{\rho}_0 = (\boldsymbol{\rho}_{01} + \boldsymbol{\rho}_{02})/2, \ \boldsymbol{\rho}_{0d} = \boldsymbol{\rho}_{01} - \boldsymbol{\rho}_{02},
\tag{11}
$$

$$
\boldsymbol{\rho} = (\boldsymbol{\rho}_1 + \boldsymbol{\rho}_2)/2, \ \boldsymbol{\rho}_d = \boldsymbol{\rho}_1 - \boldsymbol{\rho}_2.
\tag{12}
$$

where $\rho_1 \equiv (x_1, y_1)$ and $\rho_2 \equiv (x_2, y_2)$ are the position vectors of two arbitrary points in the observation plane. And we easily find from Equations (7)–(12) the following relationships:

$$
W_{tr}(\boldsymbol{\rho}_{01}, \boldsymbol{\rho}_{02}; 0) = W_{tr}(\boldsymbol{\rho}_0, \boldsymbol{\rho}_{0d}; 0)
\tag{13}
$$

$$
W_{tr}(\boldsymbol{\rho}_1, \boldsymbol{\rho}_2; z) = W_{tr}(\boldsymbol{\rho}, \boldsymbol{\rho}_d; z)
\tag{14}
$$

It is well known that the Wigner distribution function (WDF) is especially suitable for the treatment of partially coherent electromagnetic beams. The WDF of a TEEV beam

propagating through non-Kolmogorov atmospheric turbulence can be expressed in terms of the trace of the CSDM by the following formula [5,9,15–17,23,24]:

$$h_{tr}(\boldsymbol{\rho},\boldsymbol{\theta},z) = \frac{k^2}{4\pi^2}\int W_{tr}(\boldsymbol{\rho},\boldsymbol{\rho}_d;z)\exp(-ik\boldsymbol{\theta}\cdot\boldsymbol{\rho}_d)\mathrm{d}^2\boldsymbol{\rho}_d, \tag{15}$$

where, $\boldsymbol{\theta} = (\theta_x, \theta_y)$, $k\theta_x$ and $k\theta_y$ are the wave vector components along the *x*-axis and *y*-axis, respectively.

The intensity moments of the order $n_1 + n_2 + m_1 + m_2$ of the WDF in Equation (15) for the TEEV beam can be given by [5,9,15,23]

$$\langle x^{n_1}y^{n_2}\theta_x^{m_1}\theta_y^{m_2}\rangle = \frac{1}{P_0}\iint x^{n_1}y^{n_2}\theta_x^{m_1}\theta_y^{m_2}h_{tr}(\boldsymbol{\rho},\boldsymbol{\theta},z)\mathrm{d}^2\rho\mathrm{d}^2\boldsymbol{\theta}, \tag{16}$$

where the total power of a TEEV beam $P_0$ is given by

$$P_0 = \iint h_{tr}(\boldsymbol{\rho},\boldsymbol{\theta},z)\mathrm{d}^2\rho\mathrm{d}^2\boldsymbol{\theta}, \tag{17}$$

Substituting Equations (6)–(15) into Equation (16), we obtain

$$W_{LT}^2 = \langle\rho^2\rangle = \langle\rho^2\rangle_0 + 2\langle\boldsymbol{\rho}\cdot\boldsymbol{\theta}\rangle_0 z + \langle\theta^2\rangle_0 z^2 + \frac{4}{3}\pi^2 z^3 T, \tag{18}$$

where the initial second-order moments are given by

$$\langle\rho^2\rangle_0 = \langle x^2\rangle_0 + \langle y^2\rangle_0 = \frac{m+1}{4}w_{0x}^2\left(1+\beta^2\right), \tag{19}$$

$$\langle\boldsymbol{\rho}\cdot\boldsymbol{\theta}\rangle_0 = \langle x\theta_x\rangle_0 + \langle y\theta_y\rangle_0 = 0, \tag{20}$$

$$\langle\theta^2\rangle_0 = \langle\theta_x^2\rangle_0 + \langle\theta_y^2\rangle_0$$
$$= \frac{1}{k^2}\left(\frac{1}{\delta_{xx}^2}+\frac{1}{\delta_{yy}^2}\right)+\left(1+\beta^2\right)\left(\frac{m+1}{k^2\beta^2 w_{0x}^2}+\frac{m+1}{4}\mu^2 w_{0x}^2 - \frac{m\mu}{k\beta}\right), \tag{21}$$

where $\beta = w_{0y}/w_{0x}$ denotes the ellipticity and

$$T = \int_0^\infty \Phi_n(\kappa)\kappa^3\mathrm{d}\kappa, \tag{22}$$

On substituting from Equation (2) into Equation (22), the quantity *T* can be easily given by

$$T = \frac{A(\alpha)C_n^2}{2(\alpha-2)}$$
$$\times\left\{\left[2\kappa_0^2\kappa_m^{2-\alpha}+(\alpha-2)\kappa_m^{4-\alpha}\right]\exp\left(\frac{\kappa_0^2}{\kappa_m^2}\right)\Gamma\left(2-\frac{\alpha}{2},\frac{\kappa_0^2}{\kappa_m^2}\right)-2\kappa_0^{4-\alpha}\right\}, \tag{23}$$

In order to conveniently study the BW of a TEEV beam propagating through non-Kolmogorov atmospheric turbulence, the rms beam wander $B_w$ and the relative beam wander $B_{wr}$ are defined as follows

$$B_w = \left[\langle r_c^2\rangle\right]^{1/2}, \tag{24}$$

$$B_{wr} = \left[ \left\langle r_c^2 \right\rangle / W_{LT}^2 \right]^{1/2}, \tag{25}$$

where $B_w$ can represent the instantaneous distance of the TEEV beam that deviates from its beam center as it propagates in turbulence, and $B_{wr}$ can reflect the relative change.

Equations (18)–(21), (24), and (25) are the main outcomes of the present paper, which can provide the propagation rule and be a powerful tool to study and reduce the turbulence-induced beam wander of a TEEV beam propagating through non-Kolmogorov atmospheric turbulence.

## 3. Numerical Examples

In this section, we will study the effects of different beam parameters ($\lambda$, $\delta_{xx}$, $\delta_{yy}$, $\beta$, $w_{0x}$, $m$, $\mu$) and turbulence parameters ($C^2_n$, $L_0$, $l_0$, $\alpha$) on the BW of a TEEV beam propagating through non-Kolmogorov atmospheric turbulence. We have also fixed some parameters: $\lambda$ = 632.8 nm, $w_{0x}$ = 0.01 m, $\delta_{xx}$ = 8 mm, $\delta_{yy}$ = 10 mm, $\alpha$ = 3.6, $C^2_n$= 1 × 10$^{-14}$ m$^{3-\alpha}$, $l_0$ = 8 × 10$^{-3}$ m, and $L_0$ = 80 m. Unless otherwise stated, these values will be used as the calculation parameters.

Figure 1 shows the effect of the atmospheric turbulence parameters ($C^2_n$, $L_0$, $\alpha$) on rms BW and the relative BW $B_{wr}$ of a TEEV beam. We find that $B_w$ and $B_{wr}$ will become large with an increase in the propagation distance $z$. The changes are more obvious for turbulence with the large generalized structure parameter $C_{2n}$, the large generalized exponent parameter $\alpha$, and the large outer scale of turbulence $L_0$. It can be also found that the velocity of increase of $B_w$ is stable; however, $B_{wr}$ increases quickly for $0 < z < 2$ km then slowly increases up to the stable values for 2 km < $z$ < 10 km. In Figure 1(a2,b2,c2), the relative beam wander is $B_{wr} < 1$, meaning that the velocity of the BW of a TEEV beam in non-Kolmogorov atmospheric turbulence is less than that of the beam spreading.

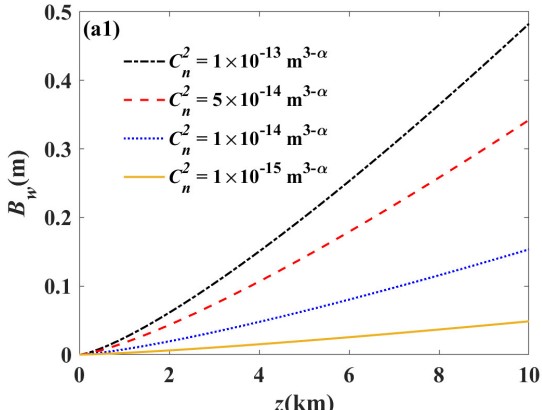
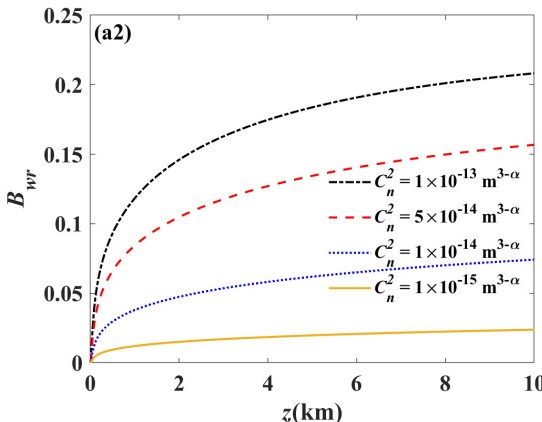

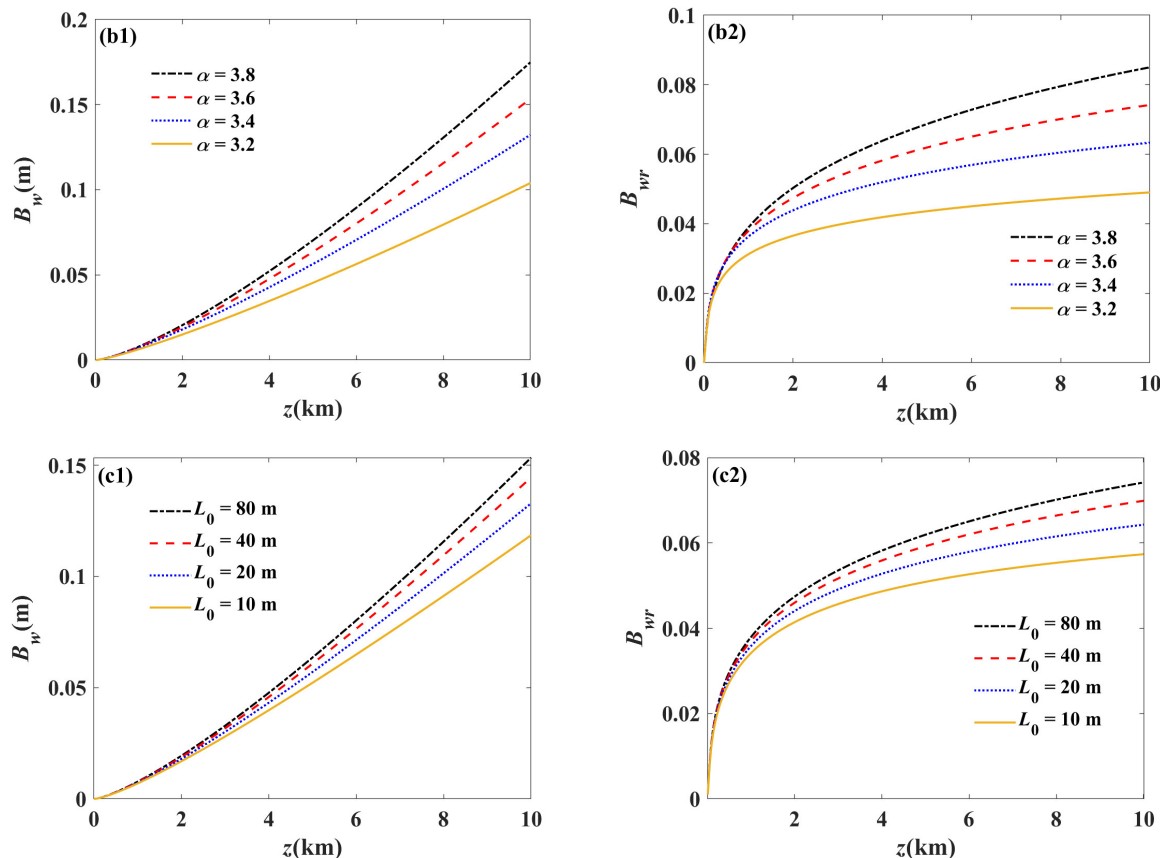

**Figure 1.** The rms beam wander $B_w$ and the relative beam wander $B_{wr}$ of a TEEV beam in non-Kolmogorov turbulence versus the propagation distance z, $\mu = -1 \times 10^{-4}$ m$^{-1}$, m = 3, $\beta$ = 0.1, (**a1,b1,c1**) denotes rms beam wander and (**a2,b2,c2**) denotes relative beam wander. (**a1,b1,c1**) show that the effect of the atmospheric turbulence parameters ($C^2_n$, $L_0$, $\alpha$) on rms BW of a TEEV beam. (**a2,b2,c2**) reveal that the effect of the turbulence parameters on the relative BW of a TEEV beam.

The effect of the ellipticity $\beta$ on the rms BW $B_w$ and the relative BW $B_{wr}$ of a TEEV beam propagating through non-Kolmogorov atmospheric turbulence is shown in Figure 2. It is seen that a TEEV beam in turbulence has a smaller $B_w$ and $B_{wr}$ than a TEEV beam with a small ellipticity $\beta$. We also find that the phenomenon is more obvious for 2 km < z < 10 km [see Figure 2a,b]. Compared with Figure 2a,c, it can be observed that the rms BW will drop with an increase of the topological charge *m*. We also find that the relative BW will reduce with a decrease of the twisted factor $\mu$ ($\mu$ < 0) in comparison with Figure 2b,d, indicating that the TEEV beam has a stronger ability to resist atmospheric turbulence by regulating jointly the twisted factor and topological charge. The reason is that the propagation of vortex beams with orbital angular momentum (OAM) in atmospheric turbulence is less affected by turbulence than non-vortex beams, and the larger the topological charge, the stronger the anti-turbulence ability, which is consistent with the results in [18,34,42–44].

Figure 3 shows the effect of the topological charge *m* on the rms BW $B_w$ and the relative BW $B_{wr}$ of a TEEV beam propagating through non-Kolmogorov atmospheric turbulence. It can be seen that the BW of the vortex beam (*m* > 0) is smaller than that of the non-vortex beam (*m* = 0); furthermore, a TEEV beam with a larger topological charge in turbulence has a smaller rms BW and relative BW. Figure 3b implies that the relative BW $B_{wr}$ quickly increases as 0 < z < 2 km, and then the change becomes slow as 2 km < z < 10 km. In Figure 3, we also find that a TEEV beam with a small value of ellipticity $\beta$ is less affected by turbulence than the circular vortex beams (ellipticity $\beta$ = 1). The reason can be explained

as the following: the elliptical vortex beams (with an ellipticity of less than 1) can increase the mutual conversion between the OAM of the twisted phase and OAM of the vortex phase in comparison with the circular vortex beams ($\beta = 1$), thereby, increasing the total OAM of the beam [32,33].

Figure 4 shows the relative BW $B_{wr}$ of a TEEV beam propagating through non-Kolmogorov atmospheric turbulence. One finds that the $B_{wr}$ of a TEEV beam increases as the twisted factor $\mu$ ($\mu > 0$) increases. However, Figure 4b–d show that the $B_{wr}$ reduces as the absolute value of the twisted factor $|\mu|$ ($\mu < 0$) increases. Compared with Figure 4a,b, it can be observed that the relative BW of a TEEV beam with a negative twisted factor $\mu$ ($\mu < 0$) is smaller than that of a TEEV beam with a positive twisted factor $\mu$ ($\mu > 0$). The reason is that the phenomenon is closely related to the handedness of the twist phase and vortex phase. When the topological charge and twist factor are all positive, the handedness of the two phases is opposite to each other, implying that the OAM of the vortex phase is offset by the twist phase. However, when the topological charge is positive and the twist factor is negative, the handedness of the two phases is the same, which means that the total OAM of the beam increases [45]. The effect of beam waist widths $w_{0x}$ and $w_{0y}$, initial transverse coherent lengths $\delta_{xx}$ and $\delta_{yy}$ on rms BW $B_w$, and the relative BW $B_{wr}$ of a TEEV beam in turbulence are shown in Figure 5. It can be found that a TEEV beam with small beam waist widths and initial transverse coherent lengths has a small rms BW $B_w$ and relative BW, indicating that a TEEV beam is less affected by atmospheric turbulence.

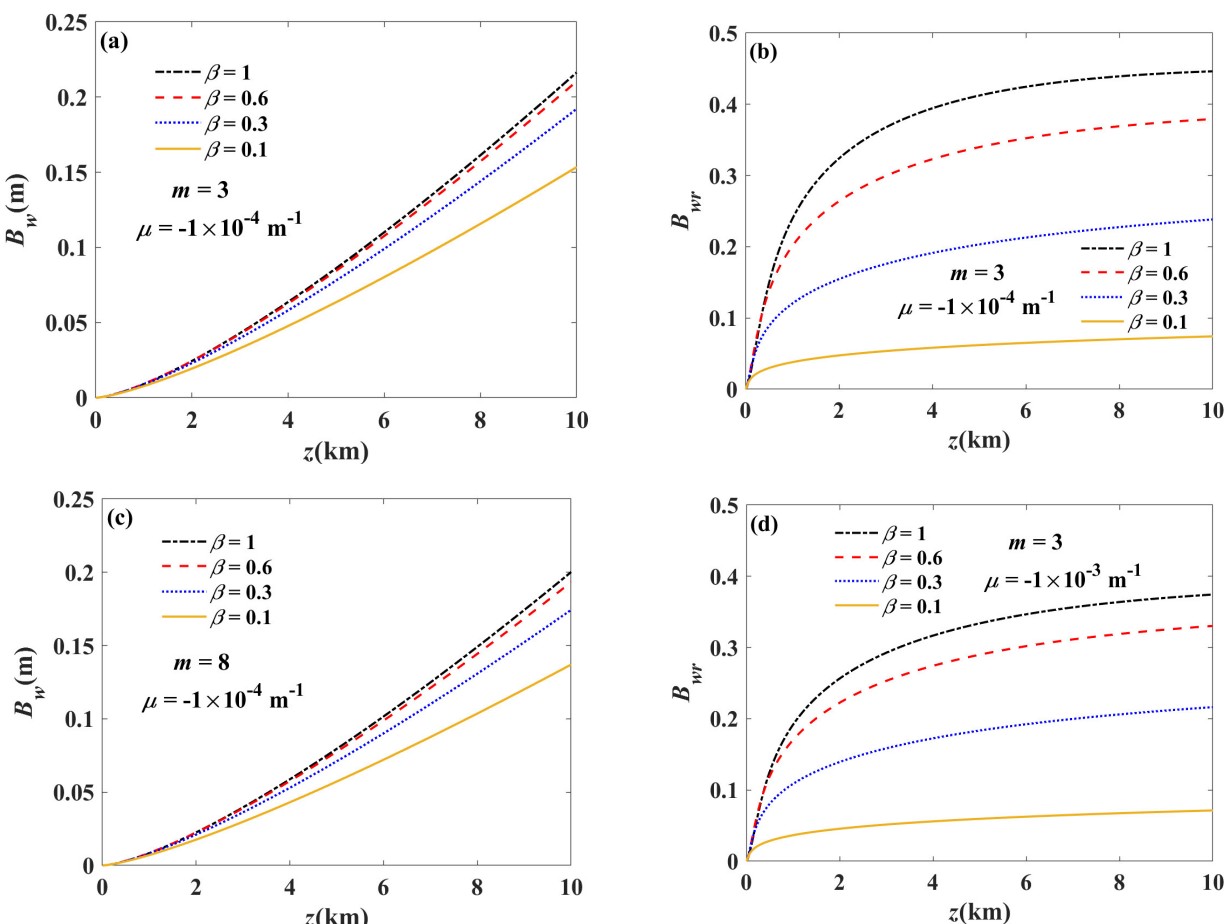

**Figure 2.** The rms beam wander Bw and the relative beam wander Bwr of a TEEV beam in non-Kolmogorov turbulence versus the propagation distance z for the different the ellipticity β, (**a,b**) μ = −1 × 10⁻⁴ m⁻¹, m = 3; (**c**) μ = −1 × 10⁻⁴ m⁻¹, m = 8; (**d**) μ = −1 × 10⁻³ m⁻¹, m = 3.

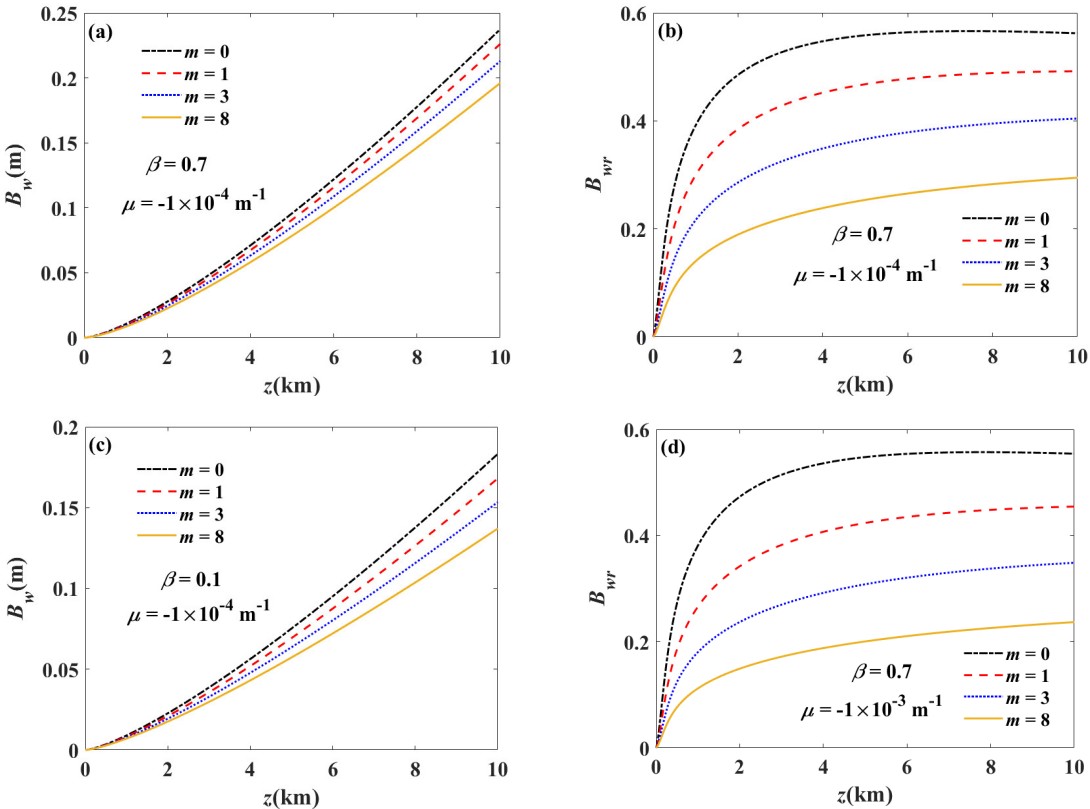

**Figure 3.** The rms beam wander $B_w$ and the relative beam wander $B_{wr}$ of a TEEV beam in non-Kolmogorov turbulence versus the propagation distance $z$ for the different topological charge $m$, (**a**,**b**) $\mu = -1 \times 10^{-4}$ m$^{-1}$, $\beta = 0.7$; (**c**) $\mu = -1 \times 10^{-4}$ m$^{-1}$, $\beta = 0.1$; (**d**) $\mu = -1 \times 10^{-3}$ m$^{-1}$, $\beta = 0.7$.

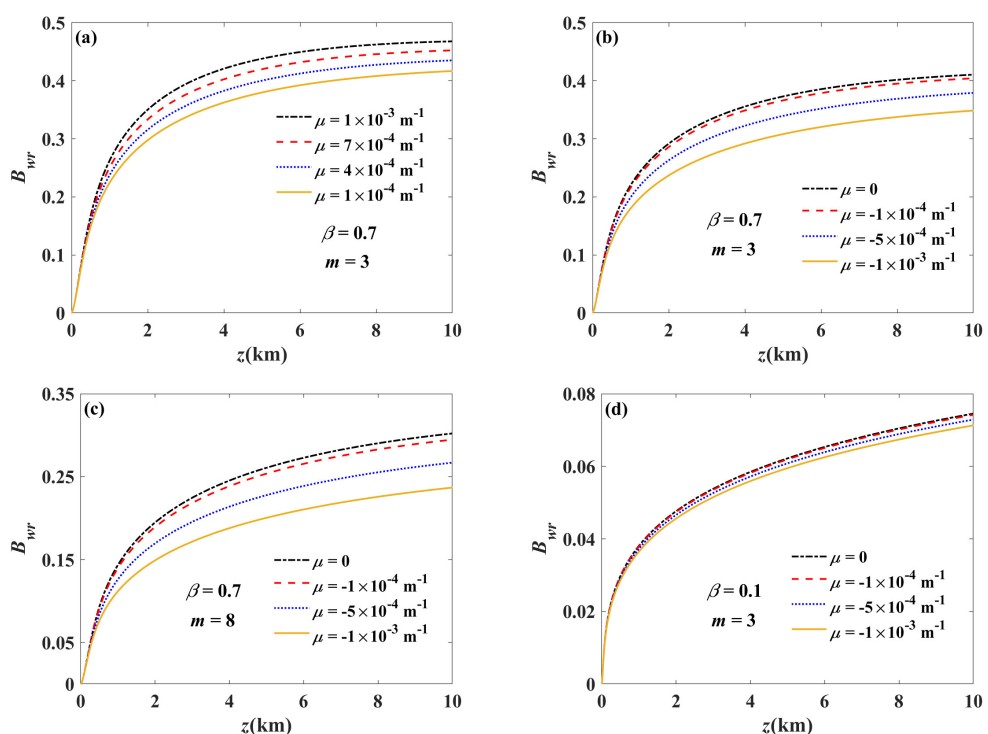

**Figure 4.** The relative beam wander $B_{wr}$ of a TEEV beam in non-Kolmogorov turbulence versus the propagation distance $z$ for the different twisted factor $\mu$, (**a**,**b**) $\beta = 0.7$, $m = 3$; (**c**) $\beta = 0.7$, $m = 8$; (**d**) $\beta = 0.1$, $m = 3$.

Figure 6 indicates the influence of the wavelength of a TEEV beam $\lambda$ on the rms BW $B_w$ and the relative BW $B_{wr}$. The longer the wavelength of the laser beam is, the smaller the $B_w$ and $B_{wr}$ are, meaning that near-infrared light has less BW than visible light in atmospheric turbulence. Figure 7 is the 3D rms beam wander $B_w$ and the relative beam wander $B_{wr}$ of a TEEV beam in non-Kolmogorov turbulence. It can be observed that the rms BW $B_w$ and the relative BW $B_{wr}$ can be reduced by decreasing beam waist widths ($w_{0x}$ and $w_{0y}$) and initial transverse coherent lengths ($\delta_{xx}$ and $\delta_{yy}$) as well as the twisted factor $\mu$ and the ellipticity $\beta$, increasing the topological charge $m$, which means $B_w$ and $B_{wr}$ are less affected by atmospheric turbulence.

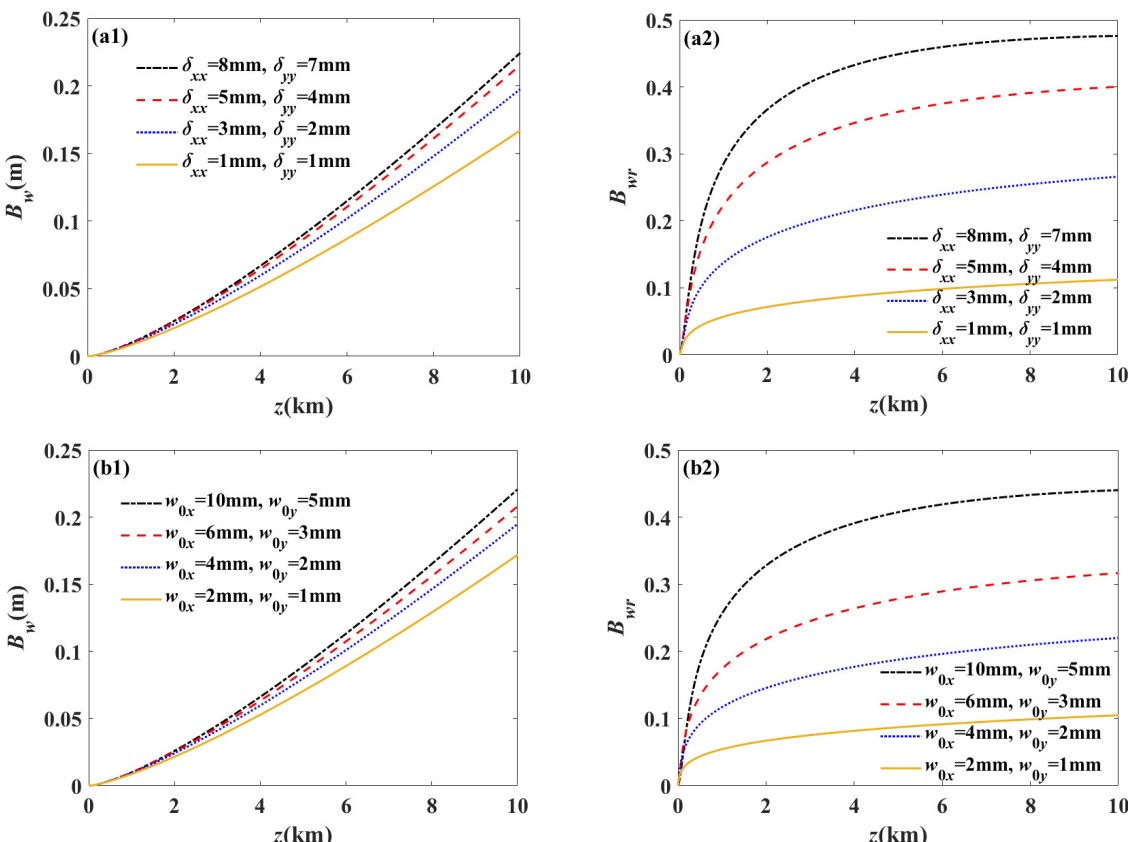

**Figure 5.** The rms beam wander $B_w$ and the relative beam wander $B_{wr}$ of a TEEV beam in non-Kolmogorov turbulence versus the propagation distance $z$ for the different beam waist widths $w_{0x}$ and $w_{0y}$, initial transverse coherent lengths $\delta_{xx}$ and $\delta_{yy}$, (**a1**,**a2**) $\mu = -1 \times 10^{-4}$ m$^{-1}$, $m = 1$, $\beta = 0.7$; (**b1**,**b2**) $\mu = -1 \times 10^{-4}$ m$^{-1}$, $m = 1$, $\beta = 0.5$.

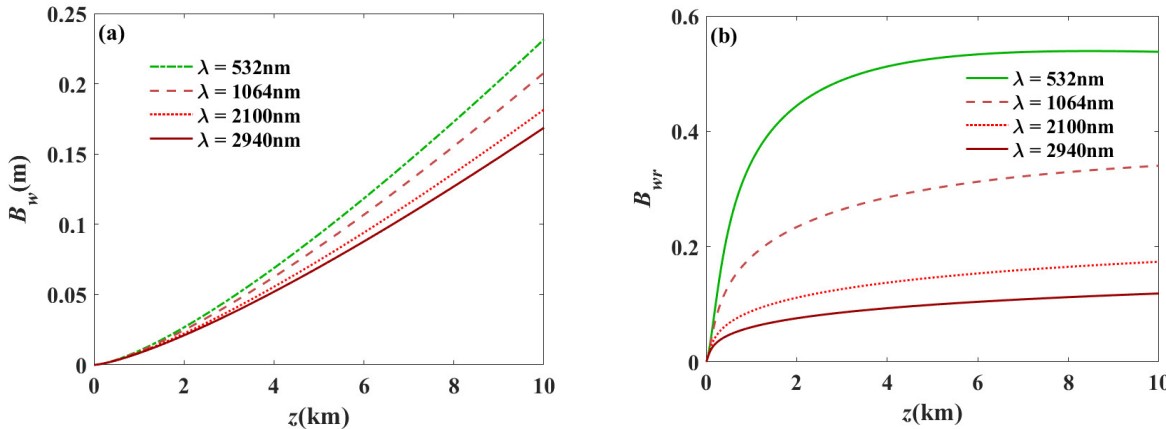

**Figure 6.** (**a**) The rms beam wander $B_w$ and (**b**) the relative beam wander $B_{wr}$ of a TEEV beam in non-Kolmogorov turbulence versus the propagation distance $z$ for the different wavelength $\lambda$, $\mu = -1 \times 10^{-4}$ m$^{-1}$, $m = 1$, $\beta = 0.7$.

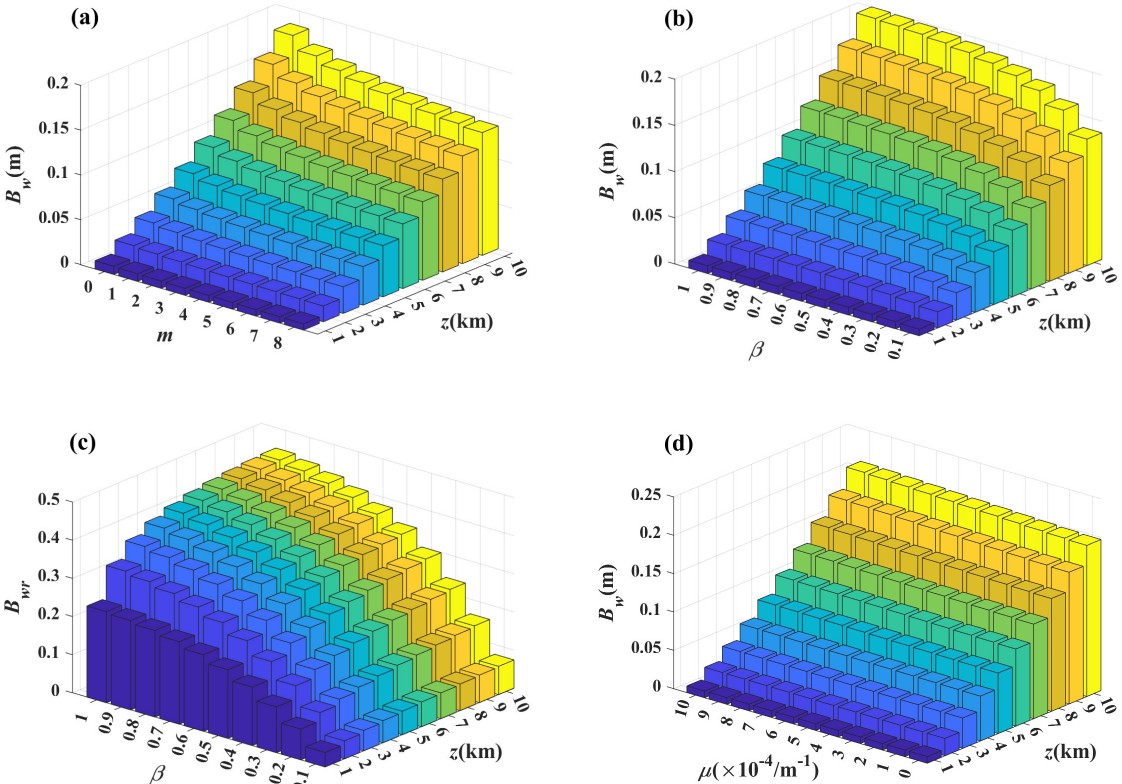

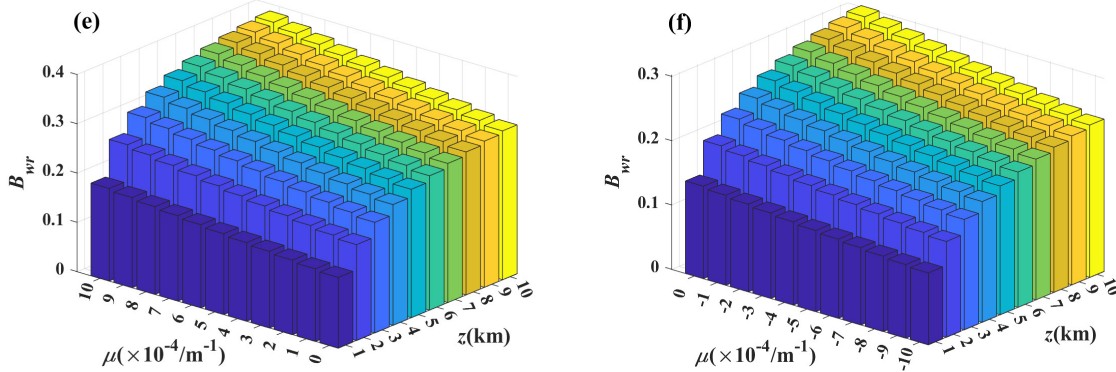

**Figure 7.** The 3D rms beam wander $B_w$ and the relative beam wander $B_{wr}$ of a TEEV beam in non-Kolmogorov turbulence (**a**) versus the propagation distance $z$ and topological charge $m$, $\beta = 0.1$, $\mu = -1 \times 10^{-4}$ m$^{-1}$, (**b**) $m = 8$, $\mu = -1 \times 10^{-4}$ m$^{-1}$, and (**c**) $m = 3$, $\mu = -1 \times 10^{-4}$ m$^{-1}$ versus the propagation distance $z$ and the ellipticity $\beta$, (**d–f**) $\beta = 0.7$, $m = 8$ versus the propagation distance $z$ and the twisted factor $\mu$.

## 4. Conclusions

We have derived the analytical expressions for the rms BW and relative BW of a TEEV beam propagating through non-Kolmogorov atmospheric turbulence with the help of the extended Huygens–Fresnel principle and the second-order moments of the WDF. Meanwhile, the numerical examples of a TEEV beam are carried out and confirm the validity of the analytical expressions. Our numerical findings demonstrate that the BW of a TEEV beam with a small ellipticity and a large topological charge as well as a small waist width and initial coherent length is less affected by turbulence, which means a TEEV beam has a stronger ability to resist atmospheric turbulence. We also find with turbulence with a larger outer scale of turbulence, the generalized exponent parameter and the generalized structure parameter on BW are more obvious. It is interesting to find that the effect of atmospheric turbulence on BW for a TEEV beam can be effectively reduced by regulating jointly the symbols and sizes of the twisted factor and topological charge. Therefore, modulation of the structure parameters ($\lambda$, $\delta_{xx}$, $\delta_{yy}$, $\beta$, $w_{0x}$, $m$, $\mu$) of a TEEV beam provides a new way to mitigate the turbulence-induced beam wander. Our work will be useful for free-space optical communications, remote sensing, and lidar distance measurement.

**Author Contributions:** Conceptualization, K.H. and Y.X.; methodology, K.H. and Y.X.; software, Y.X.; validation, K.H., Y.X., and Y.L.; formal analysis, K.H.; investigation, K.H.; resources, Y.L.; data curation, J.C.; writing—original draft preparation, K.H.; writing—review and editing, K.H.; visualization, K.H. and Y.X.; supervision, K.H.; project administration, K.H.; funding acquisition, Y.L. All authors have read and agreed to the published version of the manuscript.

**Funding:** This research was funded by National Natural Science Foundation of China (NSFC) (grant number 12033009); Yunnan Key Laboratory of Solar Physics and Space Science (grant number YNSPCC202202); Department of Science and Technology of Sichuan Province (grant number 2019YJ0470); the Sichuan Provincial University Key Laboratory of Detection and Application of Space Effect in Southwest Sichuan (grant number ZDXM202201003).

**Institutional Review Board Statement:** Not applicable.

**Informed Consent Statement:** Not applicable.

**Conflicts of Interest:** The authors declare no conflicts of interest. The funders had no role in the design of the study; in the collection, analyses, or interpretation of data; in the writing of the manuscript; or in the decision to publish the results.

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
