# Peer review of "Study of Reducing Atmospheric Turbulence-Induced Beam Wander of a Twisted Electromagnetic Elliptical Vortex Beam"

_photonics, doi:10.3390/photonics11060492_

Round 1
Reviewer 1 Report
Comments and Suggestions for Authors
Please refer to the attachment.

Reviewer 2 Report
Comments and Suggestions for Authors
In this manuscript, the authors introduced a new kind of a partially twisted vortex beam, called twisted electromagnetic elliptical vortex beam (TEEVB), and they investigated its propagation properties in non-Kolmogorov atmospheric turbulence. Based on the second-order moments of the Wigner distribution function (WDF) and non–Kolmogov spectrum for the refractive–index fluctuation, the authors obtained the analytical expressions for root-mean-square (rms) beam wander (BW) and relative BW of the TEEVB in the turbulent atmosphere. Based on the obtained formulae, they analyzed numerically the effects of the beam structure parameters, such twisted factor, topological charge, ellipticity and coherent length on the rms BW and relative BW. The paper’s purpose is to investigate a new method to mitigate the turbulence-induced BW and promote the application of the TEEVB in practical applications, such as optical communication, remote sensing, etc. The work content is relevant but the writing should be improved, there are lots of English and grammar mistakes that affect the quality of the manuscript. In overall, the proposed work provides some new results, so I think that it can be published in Photonic if the following comment points are examined in the revised version:
1-The authors should improve the readability of the paper.
2- The symbols and format in Eq.1 and Eq. 7 must be checked.
3-In Eq. 5, the term WFS is missing, why? The authors should give more details on the steps of calculating Eq. 1 to Eq. 5.
4- In Sec.4, the word “In the conclusions” is redundant, it must be omitted.
Comments on the Quality of English LanguageThe authors should improve the writing of the paper.
Reviewer 3 Report
Comments and Suggestions for Authors
In this work, the authors analytically and numerically study wander in a turbulent atmosphere of an elliptical vortex partially coherent beam (7). The authors showed that with certain parameters (topological charge, ellipticity, twisted factor) it is possible to reduce the absolute rms wander beam by approximately 1.5 times. The work can be published after the authors take into account the comments.
Comments
1. Figure 2a shows that BW decreases with increasing topological charge m of the beam (7). The reason for this effect should be explained in the text.
2. On line 173-174 it is written that “BW will reduce with decreasing the twisted factor”. This needs to be explained in the text. And why was this factor chosen negative? In equation (7), this factor can be either positive or negative.
3. The parameter β is poorly named (elliptisity), since for large ellipticity β = 0.1, and for small ellipticity β = 1. It is also necessary to explain why at large ellipticity β=0.1 BW is less than at small β=0.7 (Fig. 3).
4. In Fig. Figure 4 shows that positive values of μ result in larger BWs than negative values of μ. This needs to be explained in the text.
5. From Fig. 2a it is clear that as β decreases, BW decreases. Why does an increase in ellipticity lead to a decrease in BW? The point is not ellipticity, but an increase in the size of the beam, at least in one coordinate. Since at β=1 Wx=Wy, and at β=0.1 Wx=10Wy. Therefore, BW also decreases with increasing m, since the diameter of the beam increases with increasing m.
